# Detection and Molecular Characterization of Two Gammaherpesviruses from Pantesco Breed Donkeys during an Outbreak of Mild Respiratory Disease

**DOI:** 10.3390/v13081527

**Published:** 2021-08-02

**Authors:** Francesco Mira, Marta Canuti, Santina Di Bella, Roberto Puleio, Antonio Lavazza, Davide Lelli, Domenico Vicari, Giuseppa Purpari, Vincenza Cannella, Gabriele Chiaramonte, Giorgia Schirò, Calogero Castronovo, Annalisa Guercio

**Affiliations:** 1Istituto Zooprofilattico Sperimentale della Sicilia “A. Mirri”, Via Gino Marinuzzi n. 3, 90129 Palermo, Italy; roberto.puleio@izssicilia.it (R.P.); domenico.vicari@izssicilia.it (D.V.); giuseppa.purpari@izssicilia.it (G.P.); vincenza.cannella@izssicilia.it (V.C.); gabrielechiaramonte90@gmail.com (G.C.); giorgia.schiro91@gmail.com (G.S.); c.castronovo1992@gmail.com (C.C.); annalisa.guercio@izssicilia.it (A.G.); 2Department of Biology, Memorial University of Newfoundland, 232 Elizabeth Avenue, St. John’s, NL A1B 3X9, Canada; marta.canuti@gmail.com; 3Istituto Zooprofilattico Sperimentale della Lombardia e dell’Emilia Romagna “Bruno Ubertini”, Via Bianchi n. 9, 25124 Brescia, Italy; antonio.lavazza@izsler.it (A.L.); davide.lelli@izsler.it (D.L.)

**Keywords:** gammaherpesvirus, asinine herpesvirus, donkey, *Equid gammaherpesvirus 7*, asinine herpesvirus 5, *Percavirus*

## Abstract

Equid and asinine gammaherpesviruses (GHVs; genus *Percavirus*) are members of the *Herpesviridae* family. Though GHVs have been reported in horse populations, less studies are available on gammaherpesviral infections in donkeys. This study reports the co-infection with two GHVs in Pantesco breed donkeys, an endangered Italian donkey breed. Samples (n = 124) were collected on a breeding farm in Southern Italy from 40 donkeys, some of which were healthy or presented erosive tongue lesions and/or mild respiratory signs. Samples were analysed by using a set of nested PCRs targeting the DNA polymerase, glycoprotein B, and DNA-packaging protein genes, and sequence and phylogenetic analyses were performed. Twenty-nine donkeys (72.5%) tested positive, and the presence of *Equid gammaherpesvirus 7* and asinine herpesvirus 5 was evidenced. In 11 animals, we found evidence for co-infection with viruses from the two species. Virions with herpesvirus-like morphology were observed by electron microscopic examination, and viruses were successfully isolated in RK-13-KY cell monolayers. The histological evaluation of tongue lesions revealed moderate lympho-granulocytic infiltrates and rare eosinophilic inclusions. The detection of GHVs in this endangered asinine breed suggests the need long-life monitoring within conservation programs and reinforces the need for further investigations of GHV’s pathogenetic role in asinine species.

## 1. Introduction

The *Herpesviridae* family (order *Herpesvirales*) includes viruses infecting a wide range of animals. Among mammal species, herpesviruses have also been described in horses and, more rarely, other equids such as zebras, donkeys, and hybrids (mules and hinnies) [1,2,3].

As proposed by the International Committee on Taxonomy of Viruses (ICTV), the *Herpesviridae* family is divided into three subfamilies: *Alphaherpesvirinae*, *Betaherpesvirinae,* and *Gammaherpesvirinae* [4]. Herpesviruses (HVs) described in domestic equids are included in both *Alphaherpesvirinae* and *Gammaherpesvirinae* subfamilies.

Within the genus *Varicellovirus* of the *Alphaherpesvirinae* subfamily, viruses of the species *Equid alphaherpesvirus* 1, 3, and 4 are the most studied and described in horses, causing abortion and encephalomyelitis (EqAHV1), venereal disease (EqAHV3), and respiratory disease (EqAHV4) [2]. Other alphaherpesviruses of horses related to EqAHV-1 and EqAHV-3 (referred to as EqAHV-6 and 8) are less frequently described and affect other related domestic species such as donkeys, in which they are referred to as asinine herpesvirus 1 and 3, respectively [5,6,7].

Less studied are equid herpesviruses of the *Gammaherpesvirinae* subfamily, including *Equid gammaherpesvirus 2* and *5* (EqGHV2 and EqGHV5) (genus *Percavirus*) [8], whose natural host is the horse, and *Equid gammaherpesvirus 7* (EqGHV7), which is currently listed as a species within an unassigned genus and which has been identified in horses, donkeys, and mules (also referred to as asinine herpesvirus 2 (AsHV-2)); other yet-unclassified gammaherpesviruses (GHVs) have been identified in donkeys, for which the names asinine herpesvirus 4 (AsHV-4), asinine herpesviruses 5 (AsHV-5), and asinine herpesvirus 6 (AsHV-6) have been proposed [3,4]. Furthermore, recent studies have identified additional gammaherpesviruses of equids, but their clinical significance is still unclear [9].

Though EqGHV2 and EqGHV5 are considered endemic in horses worldwide, with their association to different illness in horses and latency in lymphoid tissues having been described [10], less documented and more unclear is their roles as pathogens of the other GHVs in other equids, especially donkeys. Indeed, GHVs have been occasionally described in donkeys since 1988 [5], either from animals without clinical signs or in those showing respiratory signs, pulmonary lesions, abortion, or neurological signs [2,11,12,13]. Overall, however, there are limited available data about herpesviruses in donkeys.

Pantesco is an ancient donkey breed imported by the Arabs [14] to the isle of Pantelleria, Sicily (Italy). For many years, Pantesco breed donkeys were employed in agriculture, used in the breeding of mules, employed in the army, and exported to other countries, but, after the Second World War, they became severely threatened by extinction [15]. In the last two decades, a protection project has been carried out to reconstitute the breed for the conservation of the Pantesco donkey population. Little is known about the health status of the Pantesco donkey and, specifically, about the potential viral pathogens affecting this small population. Therefore, a continuous monitoring is necessary to ensure their health and support their conservation programmes.

The aim of the present study was to report the detection and molecular characterization of two different GHVs in Pantesco donkeys. Phylogenetic analyses based on three different genes and virus isolation combined with electron microscopy integrated clinical and histological data to add further data to the current scientific literature.

## 2. Materials and Methods

### 2.1. Sample Collection

Samples originated from a breeding farm located in Sicily (Southern Italy) that houses pure Pantesco donkeys. The farm comprises 77 donkeys, aged from ten months to twenty-four years, living in two different and distant stables without close contact. During late September 2018, in the stable comprising 43 animals aged from ten months to twenty years, four donkeys showed clinical signs of fever, prostration, isolation attitude, loss of appetite, lacrimation, moderate dyspnoea, and cough; other animals of the stable showed lacrimation and mild respiratory signs. Following epidemiological investigations, no new introduction of donkeys to the farm or close contact with other equids during the previous months were reported. Animals in the other stable (n = 34) did not show any of the above-described clinical signs. Despite symptomatic therapies and feeding improvements, four donkeys (one yearling foal and three 9-, 11-, and 18-years old jennies) died. The foal was subjected to necropsy and the collection of tissue samples (tongue, lungs, spleen, kidney, intestine, and mesenteric lymph nodes) for further laboratory investigations.

During a follow-up visit in the stable two weeks later (mid-October), the above reported clinical signs were no more evident in most of the remaining animals (n = 39). There were, however, a few exceptions: a 19-year-old jenny and an 18-year-old jenny (id.s Francesca and Gina) showed fever, depression, and lacrimation (one of them (Francesca) also showed light erosive lesions in the edge of the tongue and in the gingival mucosal Figure 1), and six jennies and two jacks showed scarring outcomes of the lesions in the tongue and gums. Thus, blood–EDTA samples from all thirty-nine donkeys and swabs (conjunctival, nasal, and oral/tongue) from the donkeys showing erosive lesions or their scarring outcomes were collected for diagnostic purposes.

During a third visit three weeks later (7 November), blood–EDTA samples from all donkeys of the stable and swabs (conjunctival, nasal, and oral/tongue) from fourteen donkeys were further collected.

Details of collected samples are reported in Appendix A. All samples were submitted to the Istituto Zooprofilattico Sperimentale della Sicilia “A. Mirri” (Palermo, Italy) for virological and histopathological assays.

### 2.2. DNA Extraction and Pan-Herpesvirus PCR Assay

Tissue sample homogenates (tongue, lungs, spleen, kidney, intestine, and mesenteric lymph nodes) and swabs (conjunctival, nasal, and lingual) were collected and transferred to tubes containing, respectively, 9 and 3 mL of Eagle’s Minimum Essential Medium (EMEM) (Sigma–Aldrich^®^, Milan, Italy) supplemented with 2% foetal bovine serum (FBS) (EuroClone^®^, Milan, Italy) and an antibiotic–antimycotic solution (1000 U/mL penicillin G sodium salt, 1 mg/mL streptomycin sulphate, and 2.5 µg/mL amphotericin B; EuroClone^®^, Milan, Italy). Clarification was performed by centrifugation at 1.500× *g* for 10 min at 4 °C. Each sample was stored at −80 °C until use.

Viral DNA was extracted from 200 µL of organ and swab homogenates and from 100 µL of blood–EDTA samples using the DNeasy Blood and Tissue Kit (Qiagen S.p.A., Milan, Italy), according to the manufacturer’s instructions. The DNA was eluted in 200 µL of an AE elution buffer and stored at −80 °C.

To assess the presence of herpesviral DNA, all samples were tested with a pan-herpesvirus nested-PCR protocol using a set of previously described degenerate primers [16] targeting highly conserved amino acid motifs within the DNA-directed DNA polymerase (DPOL) gene.

Reactions were conducted using the commercial *Taq* PCR Core Kit (Qiagen S.p.A.), in a 50 µL reaction mix consisting of 5 µL of a 10 × PCR Buffer, 1.25 µL of a dNTP mix (200 µM), 1.2 µL of each primer (20 µM) (DFA, ILK, and KG1 for the first PCR and TGV and IYG for the nested PCR; Table 1), 0.4 µL of *Taq* DNA Polymerase, 2.5 µL of DMSO (dimethylsylphoxide, Hybri-Max^®^; Sigma-Aldrich, Milan, Italy), and 10 µL of DNA solution. Amplifications were conducted under the following thermal conditions: 94 °C for 3 min to activate TaqPol followed by 45 cycles of 94 °C for 30 s, 46 °C for 30 s, 72 °C for 60 s, and a final extension of 72 °C for 5 min.

### 2.3. Other PCRs

Samples were further analysed by two separate nested-PCRs using two sets of previously described primers [9,17] targeting the glycoprotein B (gB) and DNA-packaging protein (terminase (ter)) genes of GHV.

For the amplification of the partial gB gene, analyses were conducted using the commercial *Taq* PCR Core Kit (Qiagen S.p.A.) in a 50 µL reaction mix consisting of 5 µL of a 10x PCR Buffer, 1 µL of a dNTP mix (200 µM), 1.25 µL of each primer (0.5 µM) (2759 and, 2762as for the first PCR and 2760s and 2761as for the nested PCR; Table 1), 0.25 µL of *Taq* DNA Polymerase, and 5 or 2 µL of DNA extract for the first or nested PCR, respectively. Amplification was conducted under the following thermal conditions: 94 °C for 3 min followed by 40 cycles of 94 °C for 30 s, 46 °C for 30 s, 72 °C for 60 s, and a final extension of 72 °C for 10 min.

For the amplification of the partial ter gene, analyses were conducted using the commercial GoTaq^®^ G2 DNA Polymerase (Promega Corporation, Milan, Italy) in a 50 µL reaction mix consisting of 10 µL of a 5X GoTaq^®^ Reaction Buffer, 1 µL of a dNTP mix (200 µM), 0.6 µL of each primer (0.6 µM) (A2 and B1 for the first PCR and A3 and B2 for the nested PCR; Table 1), 0.2 µL of GoTaq^®^ G2 DNA Polymerase, and 5 or 2 µL of DNA extract for the first or nested PCR, respectively. Amplification was conducted under the following thermal conditions: 95 °C for 2 min to activate TaqPol followed by 10 touchdown thermal cycles of 95 °C for 30 s, from 68 to 50 °C for 20 s, 72 °C for 90 s, 40 cycles of 95 °C for 30 s, 46 °C for 20 s, 72 °C for 90 s, and a final extension of 72 °C for 7 min.

The products of amplification were analysed by electrophoresis on a 2% agarose gel supplemented with ethidium bromide. Positive amplicons obtained from all of the above-described PCRs were purified with Illustra™ GFX™ PCR DNA and Gel Band Purification Kit (GE Healthcare Life Sciences, Milan, Italy), and they were submitted to BMR Genomics srl (Padua, Italy) for direct Sanger sequencing.

### 2.4. Sequence and Phylogenetic Analysis

To genotype the strains under investigation, all obtained sequences were compared to each other and to reference strains present in public databases. Sequences with high-quality sequenced bases over the whole sequence length were used for phylogenetic inference. Reference sequences were identified by a BLASTn search, performed on 30 January 2021, and sequences with an identity higher than a defined cut-off (70% for gB, 80% for DPOL, and 90% for ter) with 90% sequence coverage were downloaded and used for further analyses. In total, 249 sequences from viruses of equids (163 gB, 56 DPOL, and 30 ter) were used for phylogenetic inference, while sequences of the *Caprine herpesvirus 2* (accession number: AF327834) were used as outgroups. The accession numbers of used sequences are available in the Appendix A. Alignments were performed with MAFFT [18] and the G-INS-I algorithm, and maximum likelihood phylogenetic trees [19] were constructed with MEGAX [20] using the best-fit model of nucleotide substitution identified with a model test analysis; bootstrapping (1000 replicates) was used to assess branch robustness [21]. The sequence data obtained in this study were submitted to the DDBJ/EMBL/GenBank databases under the following accession numbers: MZ209309–MZ209391.

### 2.5. Electron Microscopy (EM)

Conjunctival and nasal swabs from donkey id. Francesca were submitted to the Istituto Zooprofilattico Sperimentale della Lombardia e dell’Emilia-Romagna “Bruno Ubertini” (IZSLER, Brescia, Italy) for direct EM observation. Negative staining electron microscopy (nsEM) was performed by using the Airfuge method [22]. Conjunctival and nasal swabs were diluted in PBS and then ultracentrifuged (Airfuge, Beckman Coulter Inc. Life Sciences, Indianapolis, Indiana, USA) for 15 min at 82,000× *g* using a rotor holding six 175-μL test tubes in which specific adapters for 3 mm carbon-coated Formvar copper grids were placed. Grids were stained with 2% NaPT at pH 6.8 for 1.5 min and examined at 13,500–43,000× by using a Tecnai G2 Spirit BioTwin transmission electron microscope (FEI, Hillsboro, OR, USA) operating at 85 kV and 20,500–43,000× for at least 15 min before being considered negative. Viral particle identification was based on morphological characteristics.

### 2.6. Virus Isolation

Conjunctival and nasal swab suspensions (1 mL) from donkey id. Francesca, collected during the third visit, were inoculated on confluent RK-13-KY (rabbit kidney) monolayers in 25 cm^2^ flasks (Falcon, BD Bioscence, Basel, Switzerland). After an adsorption period at 37 °C for 1 h, the EMEM supplemented with 2% FBS and antibiotics (100 U/mL penicillin G sodium salt, 0.1 mg/mL streptomycin sulphate, and 0.25 µg/mL amphotericin B; EuroClone^®^, Milan, Italy) was added and cells were maintained at 37 °C at an atmosphere of 5% CO_2_ for 10 days with periodic microscopic examination. A blind passage was made if the cytopathic effect (CPE) was absent. Cell monolayers were frozen and thawed three times and then centrifuged at 1.500× *g* for 15 min at 4 °C; then, 1 mL of the suspension was inoculated on fresh cell monolayer. The samples were considered negative if no CPE was observed after the seventh-blind passage. In the presence of CPE, the supernatant was tested for herpesvirus DNA by PCR to confirm the presence of the infectious virus. Partial gene fragments (DPOL, GlyB, and term) of isolates were also amplified and sequenced to genotype the strains.

### 2.7. Histopathological Examination

Histopathological examination was performed on samples of the tongue with ulcerative lesions and lung of the foal. No tissues from other animals were available for this analysis. Four-micrometre-thick sections obtained from formalin-fixed paraffin-embedded tissue were set on slides treated with silane (3-aminopropyl-trieossi-silane) to avoid detachment during staining. Preparations were dried overnight in an oven at 37 °C and then dewaxed by xylene for 20 min. After a descending alcohol series (100°, 95°, 75°, and 50°), slides were washed in distilled water and then stained with haematoxylin and eosin. This was followed by the ascending scale of alcohols (50°, 75°, 95°, and 100°) and clarification in xylene. After this phase, the slides were mounted in an acrylic mounting medium (Eukitt^®^, O. Kindler, Bobingen, Germany) and used for microscopic observations.

## 3. Results

### 3.1. Necropsy

At necropsy, the foal was in a fair nutritional state; on external examination, no lesions were observed except for the leakage of foamy fluids from the nostrils, and no gross lesions were observed except for the evidence of a 2 cm rounded ulcer with moderately jagged edges in the lingual mucosa and an acute pulmonary oedema, with areas of reddening of the cranial portion and scattered petechiae of 0.5–2 cm in diameter (Figure 1).

### 3.2. PCR Detection of Herpesviruses (GHVs)

From the analysis of all samples (n = 124) collected from the 40 donkeys, 65 samples from 29 donkeys (72.5%) tested positive to the pan-herpesvirus nested-PCR targeting the DPOL gene (Appendix A). The positive donkeys comprised 6 jacks and 23 jennies with median ages of 14 (ranging from 1 to 20 years old) and 8 (ranging from 1 to 19 years old) years, respectively.

Among tissue samples collected from the dead foal, the tongue and lungs tested positive. Among live animals, 13 donkeys tested positive twice (three weeks apart), while eight donkeys tested positive both from swabs and from blood–EDTA samples at the same timepoint. Samples collected from donkeys id. Francesca, Ariel (1 year old), Gina, and Titti (8 years old) yielded higher amplification signals.

Samples were also used as input for PCRs targeting two other viral genes (gB and ter), but the nested-PCR targeting the DPOL gene yielded a higher number of amplicons. Overall, 40.5–33.3% of blood–EDTA and 90–85% of swabs samples at two different timepoints tested positive.

### 3.3. Molecular Epidemiology of Gammaherspeviruses (GHVs)

To perform molecular epidemiology, we examined a total of 59 samples collected from 23 animals at three different time points (Figure 2), from which we obtained 94 high-quality sequences. These included 34 gB, 20 ter, and 40 DPOL sequences. In all phylogenetic analyses, these sequences clustered within two separate and highly supported groups (Figure 3; Appendix A for the trees in extenso). According to the phylogenetic tree built with the DPOL sequences, for which more reference sequences are available, these two groups corresponded, respectively, to (i) the species *Equid gammaherpesvirus 7* (also known as asinine herpesvirus 2), which also included all known asinine herpesvirus 4 sequences that could be therefore considered part of this species, and (ii) a recently identified species that has not yet received official taxonomic designation and is referred to as asinine herpesvirus 5. Though similar results were obtained with the ter region, the lack of reference gB sequences made the interpretation of the results obtained with this genomic region somewhat more complicated, so groups were defined based on results obtained from the same samples using the two other genomic regions. For example, all samples from the donkey id. Gina consistently clustered within the asinine herpesvirus 5 group.

For all genes, obtained sequences of the *Equid gammaherpesvirus 7* species were more conserved (within identity: 97.2–100%) compared to those of the asinine herpesvirus 5 group (within identity: 93.3–100%), while sequences from the two species were 82–91.9% identical to each other. In *Equid gammaherpesvirus 7*, the DPOL was the most variable gene, gB was the least conserved gene for the asinine herpesvirus 5, and ter was the genetic fragment characterized by the highest identity between groups (Appendix A).

In 11 out of 23 animals, we found evidence for co-infection with viruses from the two species, although multiple samples could be obtained from just a fraction of animals and only three animals, for which more than one sample was available, consistently showed the presence of only one of the two viral species (Figure 2). In some cases (like for the donkey id. Francesca), we found one virus in samples collected at earlier time points and a different virus at later time-points, but we also found the two different viral species to be simultaneously present in the same sample collected during the same time-point, like in the case of the conjunctival swabs of the donkey id. Tosca; in the nasal swabs of the donkeys id. Titti, Iesolo, and Elvis; and in blood of the donkeys id. Ariel and Isidoro. Overall, we found evidence for *Equid gammaherpesvirus 7* in 14 (60.9%) donkeys and of asinine herpesvirus 5 in 20 animals (87%). Interestingly, asinine herpesvirus 5 was the only virus found in all swab samples collected during earlier timepoints and detected in all sample types collected at all three timepoints, while *Equid gammaherpesvirus 7* was only found in blood samples from the second timepoint and blood and swab samples from the third timepoint.

### 3.4. Electron Microscopy

Negative-staining EM showed the presence of numerous herpesviral particles in both tested swabs (conjunctival and nasal of donkey id. Francesca). Herpesvirus identification was based on the typical morphology of both the icosahedral capsids (about 100 nm in diameter), which were the most frequently observed particles, and the few pleomorphic larger enveloped particles, which appeared mostly destroyed (Figure 4).

### 3.5. Virus Isolation

Positive samples from donkey id. Francesca were inoculated onto RK-13-KY cell monolayers. All these samples showed a cytopathic effect after multiple passages, and the isolation was confirmed by PCR. Strains were isolated from the conjunctival and nasal swabs at the third- and fifth-blind passages, respectively (Figure 4). For all genes, obtained sequences belonged to the species *Equid gammaherpesvirus 7.* The analysis of sequences from the isolated virus revealed 100% nucleotide identity with sequences directly retrieved from conjunctival and nasal swabs or blood samples collected from donkey id. Francesca at the same time point. Isolates from this study were sent to be preserved and long-term stored at the Biobanca del Mediterraneo (www.bbmed.it (accessed on 23 June 2021)), a biobank for the safe storage of biological material.

### 3.6. Histopathology

Histological examination performed with haematoxylin–eosin staining revealed moderate lympho-granulocytic infiltrates affecting the lesions of the tongue and rare infiltrates including eosinophils (Figure 4). Only a marked blood congestion was evident in the lung.

## 4. Discussion

The domestication of donkeys has been dated to about 6000 years ago, and, throughout the centuries, donkeys have played significant roles in ancient states and pastoral societies [23]. Nowadays, donkeys continue to be essential in arid or rugged regions of the globe, but their utility as pack animal has been lost in developed countries, leading this species, particularly some native breeds, to extinction [15]. In Italy, six donkey breeds are already extinct. However, in the last few years, some autochthonous small populations, like the Pantesco breed, have been protected by conservation programmes [15].

This study, which was conducted on a small population of Pantesco donkeys protected by a conservation program, aimed to identify and genetically characterize two asinine GHVs.

The viruses were identified during diagnostic tests performed on a foal that died after showing mild and non-specific respiratory signs, as well as ulcers on the tongue. After this finding, animals in the same stable that showed similar respiratory signs were also tested, and a high gammaherpesviral prevalence was observed. Almost all animals spontaneously recovered within three-to-four weeks, leading to *restitutio ad integrum* and complete healing. Despite this, both blood and swabs continued to test positive. Furthermore, herpesvirus particles were directly visualized by electron microscopy from swabs collected from one donkey (id. Francesca) showing the above-described clinical signs, and the virus was isolated on cell culture. Most of the collected samples yielded detectable amplicons in both the first and nested steps, and they allowed us to obtain high-quality DPOL, gB, and ter sequences. When the amplification signal of the POL gene was weak, a weak signal or, more often, a negative result was obtained for the GlyB and, particularly, ter genes.

Both the evolution of the clinical picture and the resolution with non-specific supportive therapy led to the hypothesis that these donkeys were already harbouring a GHV infection and that the viral excretion was likely due to a reactivation of a latent infection. Though several potential stressors linked to viral reactivation were considered, we could not conclusively determine a potential cause. This evidence was corroborated by the long-term lack of introduction of other animals in the farm and of recent contact with other equids. These data suggested the circulation of GHVs in donkeys with different clinical status, ranging from healthy to moderately sick (fever, middle respiratory signs, and ulcers on the tongue), though a unique and well-defined etiopathogenetic role of GHVs cannot be confirmed in this case because the presence of other pathogens was not excluded and, therefore, needs to be further ascertained. As a note, mammalian GHVs persistently infect a wide range of animals with a periodic or continuous viral shedding. For the GHVs of equines, viral shedding has been related to different favouring factors [24], but their role in the development of specific diseases is poorly understood despite being commonly shed [25]. As for horses [25,26], the viral load assessed by quantitative PCRs could support the interpretation of clinical findings and the magnitude of GHV shedding in donkeys. Moreover, a direct relationship with clinical signs or pathological lesions should be further addressed through specific assays, such as in situ hybridization techniques, as well as by measuring the levels of antibodies against each virus, e.g., by neutralization assays.

Over the years, asinine GHVs have been predominantly reported in donkeys, but they have also been detected in healthy horses and mules [2,5,27,28,29] and in donkeys and horses showing respiratory [11,17,27,28,30,31,32], neurological [12], and abortive [13] signs. However, the evidence of GHV infection in healthy equids makes their role as pathogens questionable. Furthermore, studies based on a larger number of animals have evidenced a low viral prevalence [2,29], while other studies have reported positive [2,27,28] or negative [11,17] results in healthy donkeys from stables with past or contemporary anamnesis of respiratory signs in cohabiting equids. Further studies are therefore necessary to evaluate any clinical correlation with the different asinine GHV species, especially considering that the unambiguous causal relationship between GHV infection and disease remains a challenge in horses, in which GHVs are considered endemic [33].

In this study, the presence of GHVs in Pantesco donkeys was not unequivocally considered as causative of the observed clinical signs; however, compared to previous studies, a higher prevalence (72.5%) was observed in conjunction with these manifestations. Similarly, a higher prevalence of EqGHV-2 and EqGHV-5 was observed in horses with airway inflammation compared to clinical healthy horses [34], and the occurrence of GHV infection in healthy equids is less common compared to the prevalence of EqGHV2 and EqGHV5 in symptomatic equids [2,11,17]. More recently, the relationship between stress and nasal discharge, an increase of viral excretion, and a recrudescence of herpesviral latent infections in feral donkeys or captive Grévy’s zebras was suggested [32,35]. It is therefore possible that in our case, a respiratory infection caused herpesviral reactivation. Since the impact of these viruses remains unclear and specific control or prevention measures are not available [36], further large cohort studies could elucidate the true prevalence of the GHVs and any correlated risk factors to assess whether they are common in the asinine population and to evaluate their potential pathogenetic roles. Furthermore, the isolation of the viral strains from this asinine breed could facilitate future studies to evaluate their pathobiology or perform genomic investigations. Future in vitro studies involving different viruses will be essential to compare the biology and replication dynamics of the various herpesviral species of donkeys.

An asinine GHV was isolated for the first time three decades ago in Australia from leukocytes of an apparently healthy donkey, and it was found to be related to but divergent from the GHVs of horses, EqGHV2 and EqGHV5 [5]. This newly identified virus was defined as a “slowly cytopathic herpesvirus”, and it was classified at that time as a betaherpesvirus. Since 2002, further studies based on genomic analyses have tentatively designated the GHVs from donkeys as separate species and referred to them as AsHV-2 [2,12,13,32], AsHV-4 [11,17,28], AsHV-5 [11,17,27,28], and AsHV-6 [17]. Asinine herpesvirus 2 was isolated from the nasal secretions of mules [2], while asinine herpesvirus 5 was isolated from horses [27,29,30,31]. Later studies elucidated that many of these viruses can infect different equid species, and genetic investigations led to the official classification of asinine herpesvirus 2 as *Equid gammaherpesvirus 7*; the remaining asinine GHVs have not received an official taxonomic designation yet.

The sequences obtained in this study clustered within two separate and highly supported groups corresponding to the species *Equid gammaherpesvirus 7*, which also clearly includes all known strains of the so-called asinine herpesvirus 4, and the asinine herpesvirus 5 clade, which also includes viruses recently identified in wild Somali donkeys. Different degrees of genomic variabilities were observed between the two detected viral species, as well as among each analysed gene. The analysis was based on different targets and contributed to fill current sequence information gaps, such as by adding the glycoprotein B gene sequence of *Equid gammaherpesvirus 7*, which (to our best knowledge) was sequenced for the first time in our study, and the GlyB gene sequence of AsHV-5 from domestic equids.

Molecular analyses targeting single genes and using partial and short sequences for typing could have contributed to the literature definition of many viral groups with different given names, as previously observed [3], with some of them being redundant. These included viruses designated as asinine herpesvirus 2 or asinine herpesvirus 5, as well as viruses referred to as asinine herpesvirus 4 [11,17], asinine herpesvirus 6 [17], and untyped viruses from wild Somali asses and Damara zebras [1,37] related to equine GHVs. Our analyses showed that the EqGHV7, for which only the DPOL sequence is available, is likely the same virus as the asinine herpesvirus 4. Another example is the presence of multiple viruses in public databases referred to as “Equid herpesvirus 9” that cluster within different groups, with one of them being classifiable as asinine herpesvirus 5. Though a full genome sequence analysis is required to definitively prove relationships among strains, as observed by Kleiboeker et al. [17], the three genes investigated in this study are highly conserved within each viral species and likely sufficient for the definition of species, as in the case of the *Equid gammaherpesvirus 7* that has been defined as a species only on the basis of the polymerase gene. The sequence analysis based on all three different targets allowed for a wider comparison among the known asinine GHV sequences and filled gaps in the less studied genes.

Asinine GHVs were recovered from blood samples and conjunctival, nasal, and oral swabs, similarly to other GHVs from equines [38]. EqGHV7 and AsHV-5 can simultaneously infect the same animal or, more frequently, be identified in the same or different specimens of the same animals but in different timeframes. This occurrence was not previously reported and suggests that the GHVs described here were probably re-excreted after latency. Their evidence in different timepoints suggests that they are not constantly shed, as also observed for EqGHV2 and EqGHV5 [39,40]. Therefore, their endemism in this small donkey population could be hypothesized, and this hypothesis should be considered for long-life monitoring within conservation programmes.

To date, studies on GHV infection in asinine populations are still limited. Our data evidenced the infection of EqGHV7 and asinine herpesvirus 5 in Pantesco breed donkeys. The overall limited information on asinine GHVs and the health status of this ancient breed underlines the need for further investigations to better comprehend the epidemiology of asinine GHVs. The acquired genetic data may contribute the elucidation of the phylogenetic relationships among equid GHVs and facilitate taxonomy. The evidence of GHV infection in this endangered asinine breed reinforces the need for further investigations to better understand the pathogenetic role of these viruses in the species.

## Figures and Tables

**Figure 1 viruses-13-01527-f001:**
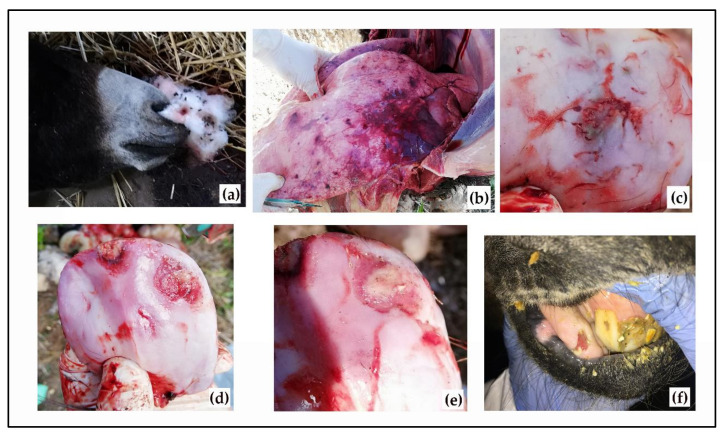
Gross lesions: leakage of foamy fluids from the nostrils (**a**), acute oedema with areas of reddening and petechiae in the lungs (**b**), and ulcerative lesions on the tongue of dead foal (**c**–**e**) and on a live animal (**f**).

**Figure 2 viruses-13-01527-f002:**
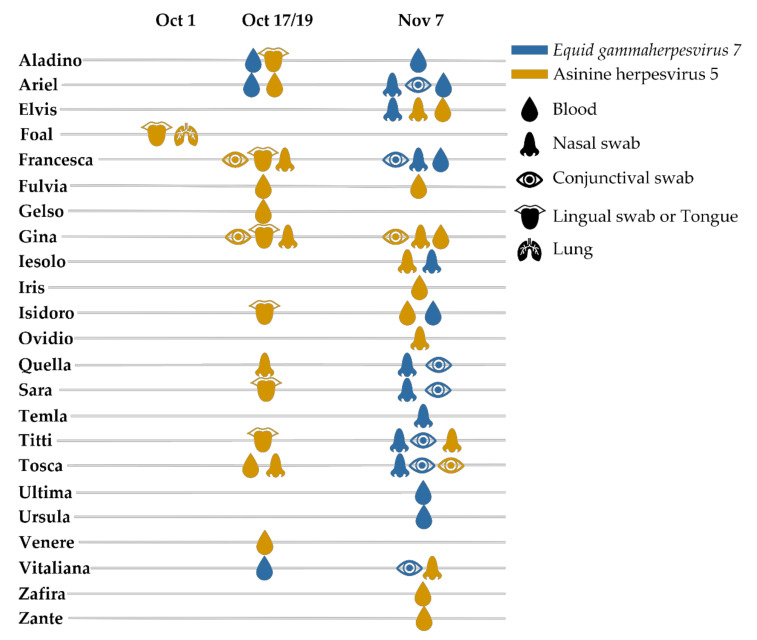
Timeline showing virus genotyping results. The results of phylogenetic-based typing of sequences obtained in this study from samples collected at three different timepoints are shown for each of the 23 positive donkeys. Sample type (blood–EDTA, tissues, or swabs) is depicted by an image that is coloured depending on the outcome of the analysis, as indicated by the legend on the right.

**Figure 3 viruses-13-01527-f003:**
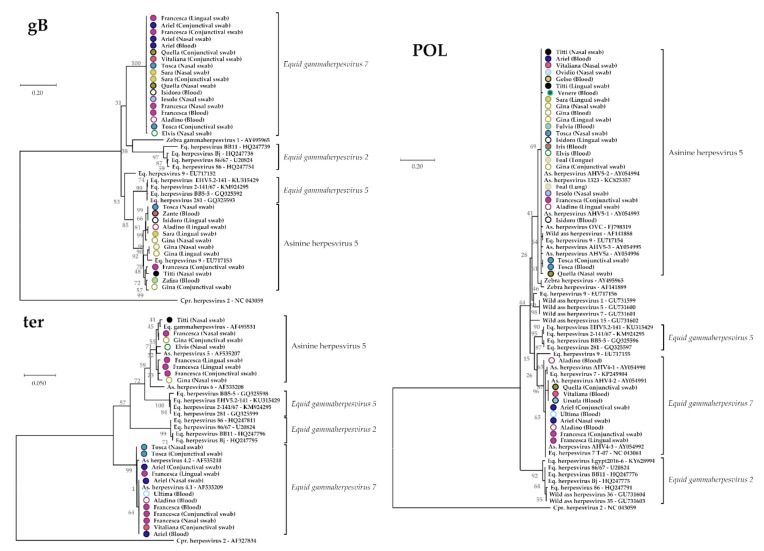
Phylogenetic trees of gammaherpesviruses identified in Pantesco breed donkeys. Maximum likelihood trees based on the glycoprotein B (gB), terminase (ter), and DNA polymerase (DPOL) genes are shown. The trees were built with MEGAX with partial nucleotide sequences (gB: 405nt; ter: 356nt; and DPOL: 182nt) using the maximum likelihood approach with Kimura 2 parameter (gB) or Tamura 3 parameter (DPOL and ter) models, identified as the best fitting models after using the model test analysis implemented in MEGAX. A discrete gamma distribution was used to model evolutionary rate differences among sites (+G = 0.3522 for gB, +G = 0.4073 for DPOL, and +G = 0.8178 for ter) and branch robustness was evaluated through 1000 bootstrap replicates. Official (italics) or provisory species names are indicated when available, and sequences obtained in this study are colour-coded, with each colour representing one donkey whose id. (and type of sample) are indicated.

**Figure 4 viruses-13-01527-f004:**
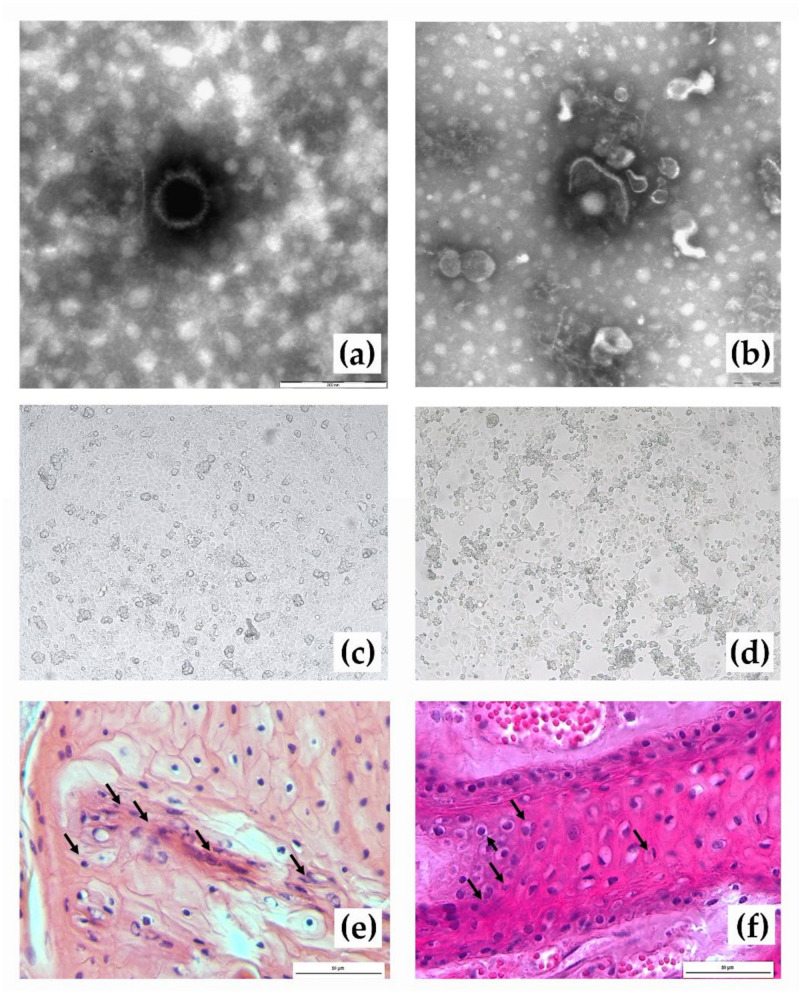
Electron micrograph of an isolated capsid of herpesvirus virion (bar = 200nm) (**a**) and of a full particle showing the envelope partially fragmented (bar 300 nm) (**b**) 2% NaPT staining; virus isolation on RK-13-KY cells with non-infected RK-13-KY cell monolayer (control) (**c**) and cytopathic effect observed in RK-13-KY cells infected with *Equid gammaherpesvirus 7* at 100× magnification (**d**); the haematoxylin and eosin staining of the epithelium shows lympho-granulocyte inflammatory infiltrates (**e**). Eosinophilic inclusions (**f**) are observed in some cells. Images were taken at 40x magnification, and the bar represents 50 µm.

**Table 1 viruses-13-01527-t001:** Primers used for the amplification and sequencing of the gammaherpesvirus partial genes.

Target	Assay	Sense	Primer	Sequence (5′–3′)	Position ^a^	Amplicon Size (bp)	Reference
DNA polymerase (DPOL) gene	First	forward	DFA	GAYTTYGCNAGYYTNTAYCC	1744–1763	787	[16]
		forward	ILK	TCCTGGACAAGCAGCARNYSGCNMTNAA	2006–2033		
		reverse	KG1	GTCTTGCTCACCAGNTCNACNCCYTT	2446–2471		
	Nested	forward	TGV	TGTAACTCGGTGTAYGGNTTYACNGGNGT	2041–2069	218	
		reverse	IYG	CACAGAGTCCGTRTCNCCRTADAT	2236–2259		
Glycoprotein B (GlyB) gene	First	forward	2759s	CCTCCCAGGTTCARTWYGCMTAYGA	1358–1382	645	[9]
		reverse	2762as	CCGTTGAGGTTCTGAGTGTARTARTTRTAYTC	1974–2003		
	Nested	forward	2760s	AAGATCAACCCCACNAGNGTNATG	1480–1503	508	
		reverse	2761as	GTGTAGTAGTTGTACTCCCTRAACATNGTYTC	1957–1988		
DNA-packaging protein (ter) gene	First	forward	A2	TTGTGGACGAGRSNMAYTTYAT	947–968	525	[17]
		reverse	B1	ACAGCCACGCCNGTNCCNGANGC	1450–1472		
	Nested	forward	A3	GCAAGATCATNTTYRTNTCNTC	1019–1040	425	
		reverse	B2	TGTTGGTCGTRWANGCNGGRTC	1423–1444		

^a^ Nucleotide positions refer to the prototype Equid herpesvirus 2, strain 86/67 (Australia, 1967—RefSeq sequence: NC_001650.2).

## Data Availability

Sequence data were submitted to the DDBJ/EMBL/GenBank databases under accession numbers MZ209309-MZ209391.

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
