# Peer review of "Detection and Molecular Characterization of Two Gammaherpesviruses from Pantesco Breed Donkeys during an Outbreak of Mild Respiratory Disease"

_viruses, 2021, doi:10.3390/v13081527_

Round 1

Reviewer 1 Report

The research article submitted by Mira et al reports the detection and molecular characterization of two different GHVs (Equid gammaherpesvirus 7 and asinine herpesvirus 5) -in Pantesco breed donkeys. The article gets the attention as it is highly necessary for continuously monitoring GHV infection in endangered breed for conservation strategies. The article is well written in all sections. However, the manuscript needs more data and the result section needs to elaborate.

Places that needs attention 

i) Partial gene fragments of isolates in RK-13-KY from Francesco was sequenced. However, it’s worth mentioning if any changes in amino acids were seen in the isolates as compared to the sequences retrieved from tissues or blood samples.

ii) Have you guys tried the virus isolation from all samples other than Francesco. If so, mention from how many positive samples, virus was cultured and sequenced. This tells which virus is adaptive over the other, in case of coinfection.

iii) Why the virus isolation/EM/histopathology was limited to Francesco. It seems Ariel, Gina and Titti also had higher amplification signal and virus isolation won’t be difficult. It would be great if the histopathology from Gina (which has only asinine herpesvirus 5 infection), can be compared to the one having co-infection like in Francesco.

iv) Fig 4: Please mark with arrows what you are describing. For eg: e and f –give emphasis on lympho granulocyte inflammatory infiltration with arrows. Show eosinophilic inclusions in some different colored arrows.

v)The histopathology from all animals if done is desirable and can be scored and presented in a graphical way.

Author Response

Rebuttal Letter

Manuscript ID: viruses-1290452

Journal: Viruses

Title: Detection and molecular characterization of two gammaherpesviruses from Pantesco breed donkeys during an outbreak of mild respiratory disease 

Dear Editors,

on behalf of the Authors of the manuscript, I express our cordial gratitude to the Reviewers for the valuable and constructive review of our paper. The reviewers’ comments helped us to further improve the overall quality of the manuscript. In this revision, we included all the suggestions and corrections as proposed by the reviewers. We also made minimal further correction to improve specific parts of the manuscript. Point-by-point responses to each reviewers’ comments are listed below.

Sincerely, Dr. Francesco Mira and Dr. Santina Di Bella.

Reviewer #1:

The research article submitted by Mira et al reports the detection and molecular characterization of two different GHVs (Equid gammaherpesvirus 7 and asinine herpesvirus 5) -in Pantesco breed donkeys. The article gets the attention as it is highly necessary for continuously monitoring GHV infection in endangered breed for conservation strategies. The article is well written in all sections. However, the manuscript needs more data and the result section needs to elaborate.

We thank the Reviewer for the positive global comment and the constructive suggestions. We have followed the Reviewer’s advice and revised the manuscript accordingly. Below the point-by-point responses to each comment.  

Places that needs attention

  1. i) Partial gene fragments of isolates in RK-13-KY from Francesco was sequenced. However, it’s worth mentioning if any changes in amino acids were seen in the isolates as compared to the sequences retrieved from tissues or blood samples.

All three partial gene fragments sequenced from isolates in RK-13-KY from donkey id. Francesca were compared to those retrieved from conjunctival and nasal swabs or blood samples collected at the third time point. The sequence analysis revealed 100% nucleotide identity and, thus, no change in the amino acid sequences was observed. These data were included in the revised version of the manuscript at lines 203-204 (par.2.6) and 318-321 (par.3.5).  

  1. ii) Have you guys tried the virus isolation from all samples other than Francesco. If so, mention from how many positive samples, virus was cultured and sequenced. This tells which virus is adaptive over the other, in case of coinfection.

In this study we focused our attention on samples from Francesca because of the overall clinical signs and pathological lesions. Thus, direct electron microscopy observation was performed, showing the presence of herpesviral particles in swab samples. Therefore, virus isolation from two samples was attempted. Based on the results of this study, we agree with the Reviewer that attempting virus isolation from other positive samples would give important information about viral replication and competition between strains in case of co-infection. However, determining in vitro replication dynamics was not initially included in the scope of our investigation, which was mainly focused on a diagnostic approach, but we will certainly consider this indication for a follow-up study. This necessity has been highlighted in the text (at lines 405-407).

iii) Why the virus isolation/EM/histopathology was limited to Francesco. It seems Ariel, Gina and Titti also had higher amplification signal and virus isolation won’t be difficult. It would be great if the histopathology from Gina (which has only asinine herpesvirus 5 infection), can be compared to the one having co-infection like in Francesco.

We agree with the reviewer that these investigations would be important to compare the infection caused by the two viral species.  However, tissue samples were only, and unfortunately, available for just one dead donkey (see below) and a larger scale histological investigation was therefore not possible to perform. A follow-up study focused on differences between the 2 viral species, including in-vitro replication, EM, and histology, would give important information about both viruses and add further useful data. We have further specified in the manuscript the lack of tissues from other animals (lines 219-220 and 405-07) and the need of performing more in vitro characterization studies.

  1. iv) Fig 4: Please mark with arrows what you are describing. For eg: e and f –give emphasis on lympho granulocyte inflammatory infiltration with arrows. Show eosinophilic inclusions in some different colored arrows.

As suggested, lympho granulocyte inflammatory infiltration and eosinophilic inclusions were marked with arrows in figure 4 and caption was reviewed accordingly (lines 333-334).

v)The histopathology from all animals if done is desirable and can be scored and presented in a graphical way.

During the early stages, four donkeys died but, unfortunately, only one animal was subjected to necropsy and tissue collection for histopathological examination. Therefore, the results of this examination cannot be compared with other dead donkeys. For the remaining animals, the evaluations were based only on clinical examinations, without specific pathological assays. Therefore, although this is a very interesting aspect to evaluate, the request for scoring and comparing histopathological results cannot be complied. We have further specified the lack of tissues from other animals in the manuscript (lines 219-220).

Reviewer 2 Report

the manuscript reports the detection and co-infection of Pantesco breed donkeys, an endangered Italian donkey breed, with two GHVs; Equid gammaherpesvirus 7 and asinine herpesvirus 5. Samples were analysed by using a set of nested PCRs targeting the DNA polymerase, the glycoprotein B, and the DNA-packaging protein genes. Furthermore, one virus was isolated on cell culture and virus particles were directly detected by EM.    

the manuscript is well written and results are well presented. This study will add to the efforts to protect endangered animals while maintained in captivity.

1- while it might be clear that the occurrence of such infection is most likely due to reactivation, I wonder what could be the stress conditions that could facilitate such reactivation; please comment. In this regard, are there any other animal species that are are close to this farm? could it be that the virus jumped from another species?

2- Isolating the virus is an important step for future studies. I wonder why the authors did not try to fully sequence the genome (giving the advancement in NGS technology) to add more insight on the isolated virus instead of sequencing small portions of some genes?

3- The authors have used a complicated abbreviation for virus names; e.g. EqAHV1, EqGHV2. Although clear, mostly in the literature (as well as well-known in this field) it is preferred to use the simple (classical) abbreviation (EHV-1, EHV-2). In line, 387, the authors used EHV-2 and EHV-5 instead of their abbreviation that they used all over the manuscript. Using the (classical) abbreviation will facilitate the easy searching process in the future.

Author Response

Rebuttal Letter

Manuscript ID: viruses-1290452

Journal: Viruses

Title: Detection and molecular characterization of two gammaherpesviruses from Pantesco breed donkeys during an outbreak of mild respiratory disease 

Dear Editors,

on behalf of the Authors of the manuscript, I express our cordial gratitude to the Reviewers for the valuable and constructive review of our paper. The reviewers’ comments helped us to further improve the overall quality of the manuscript. In this revision, we included all the suggestions and corrections as proposed by the reviewers. We also made minimal further correction to improve specific parts of the manuscript. Point-by-point responses to each reviewers’ comments are listed below.

Sincerely, Dr. Francesco Mira and Dr. Santina Di Bella.

Reviewer #2:

the manuscript reports the detection and co-infection of Pantesco breed donkeys, an endangered Italian donkey breed, with two GHVs; Equid gammaherpesvirus 7 and asinine herpesvirus 5. Samples were analysed by using a set of nested PCRs targeting the DNA polymerase, the glycoprotein B, and the DNA-packaging protein genes. Furthermore, one virus was isolated on cell culture and virus particles were directly detected by EM.   

the manuscript is well written and results are well presented. This study will add to the efforts to protect endangered animals while maintained in captivity.

We thank the Reviewer for the positive global comment and the constructive suggestions. We agree with the Reviewer’s comments, and we revised the manuscript accordingly. Below, all the point-by-point responses.  

1- while it might be clear that the occurrence of such infection is most likely due to reactivation, I wonder what could be the stress conditions that could facilitate such reactivation; please comment. In this regard, are there any other animal species that are are close to this farm? could it be that the virus jumped from another species?

The potential stress conditions were only hypothesized but not confirmed. Indeed, both toxic substances or the need of feeding improvements and some medical treatment could have potentially facilitated the virus reactivation but any clear correlation with any of these factors could not be elucidated. Moreover, this stable is placed in an ample forest area, distant from the urban area. So, other domestic equids were not housed close to this farm, and information on wild species eventually present in the same area were not reported from public or private veterinary practitioners. For these reasons we suggested the potential reactivation rather than other sources of infection, but we did not have enough elements to speculate on the type of stressor that caused it. Additionally, since we could not conclude that herpesviruses were the sole cause of the respiratory symptoms, it is possible that a concomitant respiratory infection by any other infectious agent was the stress factor that caused viral reactivation. These concepts have now been added to the manuscript at lines 362-363 and 399-400.

2- Isolating the virus is an important step for future studies. I wonder why the authors did not try to fully sequence the genome (giving the advancement in NGS technology) to add more insight on the isolated virus instead of sequencing small portions of some genes?

Partial gene sequencing was performed to confirm the presence of infectious virus in the isolates and to genotype the strains. We agree that the whole genome sequencing could add more insights on the isolated virus and a deep genome sequencing by using NGS is useful for these purposes. However, the full genetic characterization of the two viruses was outside the scope of this investigation and complete genome sequencing coupled with in-depth genomic characterization will be the subject of future work.  

3- The authors have used a complicated abbreviation for virus names; e.g. EqAHV1, EqGHV2. Although clear, mostly in the literature (as well as well-known in this field) it is preferred to use the simple (classical) abbreviation (EHV-1, EHV-2). In line, 387, the authors used EHV-2 and EHV-5 instead of their abbreviation that they used all over the manuscript. Using the (classical) abbreviation will facilitate the easy searching process in the future.

We thank the reviewer for this comment. Indeed, in literature viral nomenclature that uses simple abbreviations is widespread. However, there is a great confusion about virus naming (e.g. different viruses with the same name and one virus with multiple names). Therefore, we have decided to follow what indicated in the latest ICTV reports, where it is used a 2 letter-code identifying the host, followed by 3 letters indicating the subfamily, followed by a number. More information can be found at:

https://talk.ictvonline.org/files/ictv_official_taxonomy_updates_since_the_8th_report/m/animal-dna-viruses-and-retroviruses/11017

Indeed, at line 387 (at line 393 of the revised manuscript) we used a different nomenclature, and thus it has been fixed.

Round 2

Reviewer 1 Report

Thank you for addressing the comments and modifying the manuscript.